# The Osteosynthesis of the Mandibular Head, Does the Way the Screws Are Positioned Matter?

**DOI:** 10.3390/jcm11072031

**Published:** 2022-04-05

**Authors:** Marcin Kozakiewicz, Izabela Gabryelczak

**Affiliations:** Department of Maxillofacial Surgery, Medical University of Lodz, 113 Żeromskiego Str., 90-549 Lodz, Poland; gabryelczakizabeal@gmail.com

**Keywords:** condylar head fracture, surgical treatment, open rigid internal fixation, headless screw, positional screws, titanium, magnesium, mandible fracture, long-term results, computer tomography

## Abstract

Currently, an increasing number of medical centers are treating mandibular head fractures surgically. Dedicated screws for compression osteosynthesis have been developed. However, due to the very limited size of the fractured bones and the considerable technical difficulties accompanying the execution of the fixation, there is little room for correction of the positioning and reinsertion of the screws. Therefore, knowing the optimal position of the fixation material is crucial for therapeutic success. The aim of this study is the evaluation of fixation screw position on the mandibular ramus height obtained in the treatment of the condylar head fracture. A total of 57 patients were included in this study. The loss of mandibular ramus height on computed tomography twelve months after mandibular head osteosynthesis was evaluated in relation to the initial distance of the screws from the fracture line, the angle of insertion of the screw into the bone, and the size of the protrusion to the inner side of the condyle. The relationship of the proximity of the screw to the fracture line, angulation, and the size of the protrusion with the loss of ramus height was confirmed. Conclusions: the optimal location for the superior screw is approx. 4 mm below the fracture line (with any angulation), inferior screw is approx. 8 mm (with any angulation), and anterior screw position is approx. 4–5 mm distant from fracture line with the best angulation of 130 degrees to the lateral mandible ramus surface in the coronal plane.

## 1. Introduction

Facial injuries are frequent trauma and most common in males (56.8–92.8%) [1,2,3,4], and the mean age ranges from 24 to 51 years [2,5]. Condylar fractures are one of the most frequent of mandibular fractures: 21–52% [6,7,8]. It appears that fractures of the mandibular head are not as rare as previously reported, e.g., the 11 fractures in 40 described condylar fractures [9]. The growing interest in surgical treatment [10,11,12,13] and free availability of computed tomography [14,15] stimulates an increase in the diagnosis of mandibular head fractures. A secondary benefit of this development is the ability to accurately monitor distant treatment outcomes in terms of the preservation of anatomical structures (Figure 1). Since the presentation by Kermer, Undt, and Rasse [16], the technique for the use of long screws in osteosynthesis of mandibular heads notes the current stage of medical development in this field.

Open treatment involves extracting a small fragment (proximal fragment) of the mandibular head from the infratemporal fossa using raspators. The reduction of it is conducted with good visibility of the posterior surface of the mandibular head, followed by fixation with the larger fragment (distal fragment). This fixation can be performed with long screws of different constructions [17,18,19,20]. It seems that the best current compromise between the number of screws to be used, the size of the mandibular head, and the needs for stabilization is the use of three screws [21]. There will then be a good chance of rigid fixation. Thus, the surgeon has three screw insertion sites, three insertion angles to choose from, and three screw lengths to use to achieve the ORIF, i.e., open reduction and rigid fixation.

The aim of this study is the evaluation of the fixation screw position on the mandibular ramus height obtained in treatment of the condylar head fracture.

## 2. Materials and Methods

Patients were admitted as part of the in-patient emergency service, transferred from other hospitals, or the patients themselves reported to the outpatient clinic. After assessment of the general condition and possible respiratory and circulatory stabilization, treatment with maxillofacial surgery was planned. All patients had taken a multislice computed tomography (CT). Mandible head fracture classifications according to Neff were applied (Figure 2) [19].

Inclusion criteria: recent trauma, condylar head fracture type B or C, fragment angulation > 10°, fragment overlapping > 2 mm, preauricular approach, and complete radiological documentation. Patients operated upon on Tuesdays and Thursdays received titanium fixation, while patients operated upon on Wednesdays and Fridays received magnesium fixation (the choice of fixation material was decided on the day the patient arrived at the hospital). Exclusion criteria: type A head fractures, fragment angulation ≤ 10°, fragment overlapping ≤ 2 mm, close (i.e., conservative) treatment, and failure of the patient to report for follow-up examinations.

The clinical material consisted of 39 patients with single fractures of the mandibular head and 18 patients with bilateral fractures, making a total of 74 mandibular head fractures. Type B represented 19 fractures and type C represented 55 fractures. The study group was predominantly male (61) and the age of the patients ranged from 12 to 72 years (mean was 37 ± 18 years). There were 13 smoking patients and 22 internal medicine patients. Nicotinism was related to the existence of internal medicine diseases in patients.

In the patients studied, there were 18 cases with dental fractures; 33 cases were accompanied by a mandibular body fracture, 12 cases were accompanied by a zygomatic bone fracture, and 13 cases had other distant injuries (spinal fracture, femur fracture, or lung contusion).

Post-traumatic shortening of the mandibular ramus (i.e., overlapping of the fragments) of 8.1 ± 4.5 mm was noted. The deviation of the proximal (small) fragment from the axis of the mandibular condylar process was 37° ± 22°. The number of free mandibular head fragments (except the distal, i.e., lower, ramus fragment) was 2.3 ± 1.1. Dislocation in the temporomandibular joint was observed in 62 cases and in the remaining 12 there was only displacement. Delay to treatment was 6.8 ± 8.8 day.

Patients were operated in antibiotic prophylaxis under general anesthesia with intubation through the nose, floor of the mouth, or tracheostomy. In cases of concomitant mandibular fractures, mandibular head fractures were fixated first from the preauricular approach. Efforts were made to insert 3 screws each time for the osteosynthesis of one mandibular head. Headless and solid compression screws made of titanium alloy and magnesium alloy (ChM, Juchnowiec Kościelny, Poland) were used. The lengths of the screws used were 14 mm, 16 mm, or 18 mm and the system was: 1.8 or 2.2 (Figure 3).

An attempt was made to fix the fracture with three screws: the first in the superior position (S), the second in the inferior position (I), and the third in the anterior position (A) in relation to the other two (Figure 4). All screws were directed with the tip to the medial pole of the mandibular head. The superior screw was used for all 74 osteosyntheses. The screw in the inferior position was applied in 70 osteosyntheses. The screw in the anterior position was used in only 16 fixations. Thus, most patients were operated with a 2-screw fixation (2.16 ± 0.57 screws per fixation). The wound was closed in layers without a drain. The correctness of the anatomical reduction of bone fragments was checked in the computer tomography examination on day 1 post-operationally (Figure 4). The tangent line (T) was determined in the side view of the mandibular ramus to determine the distance of the screw edge from the fracture line. This distance was measured tangentially to the T line for each screw. The angle of the screw to the outer surface of the mandibular condyle was then measured for each screw. Postoperative CT scans in frontal (coronal) sections were used for the measurements. Angle arms are the length of the screw to the length of the mandibular ramus. The technique for measuring the insertion angle of the screws into the mandibular head is shown in the image below (the right panel in Figure 3). The amount of screw protrusion on the internal side of the mandibular condylar process was also checked on the coronary sections. The tests were performed at RadiAnt (Medixant, Poznań, Poland), setting the window level to 300 HU and the window width to 1500 HU (bone window). Next, a computed tomography scan was acquired after 12 months. The height of the mandibular ramus was then measured on it; it was checked whether the ramus had decreased in relation to the computed tomography performed immediately after the surgery. The measurement of the height of the mandibular ramus was made parallel to the T line from the highest point of the mandibular head to the lowest point of the mandibular angle [22].

The structure of the bone union was also examined by means of digital texture analysis on coronary sections of the mandibular head [23]. ROI covered the post-fracture site and the next controlled ROI in the cancellous bone of mandibular head were described in MaZda 4.6 software developed by University of Technology in Lodz, Poland [24]. The Bone Index (BI) texture feature in ROIs was calculated for the control bone and for the post-fracture site:(1)$$Bone\;Index=\frac{(-\sum_{i=1}^{Ng}p_{x-y}\left(i\right)log\left(p_{x-y}\left(i\right)\right))\sum_{i=1}^{Ng}\sum_{k=1}^{Nr}p\left(i,k\right)}{\sum_{i=1}^{Ng}\sum_{k=1}^{Nr}k^{2}p\left(i,k\right)}$$
where *Nr* is the number of series of pixels with density level *i* and length *k*; *Ng*—number of levels for image optical density; *p* is probability, *log* is common logarithm [25]; *Nr*—number of pixels in series; [26,27]. The equation above was used for the Bone Index construction [28], which represents the ratio of the measure of the diversity of the structure observed in the radiograph to the measure of the presence of uniform longitudinal structures (Figure 5) [22,23].

The statistical analysis includes the feature distribution evaluation, mean (*t*-test) or median (W-test) comparison, χ^2^ tests of independence, analysis of regression, and one-way analysis of variance. Detected relationships were assumed to be statistically significant when *p* < 0.05. Statgraphics Centurion version 18.1.12 (StatPoint Technologies, Warrenton, VA, USA) was used for statistical analyses.

## 3. Results

The mean values of the screw distance from the fracture line, the screw insertion angle in relation to the lateral plane of the mandibular ramus, and the extent of penetration to the internal surface of the condylar process obtained for the study group are shown in Table 1. The loss of mandibular ramus height observed 12 months after treatment is related to the distance of the superior screw from the fracture line (correlation coefficient CC = −0.23, R^2^ = 5%, *p* < 0.05). A similar relationship exists for the inferior screw (CC = 0.31, R^2^ = 10%, *p* < 0.05) and the anterior screw (CC = 0.71, R^2^ = 50%, *p* < 0.05), as shown in Figure 6.

When evaluating the screw insertion angle, there was no relationship of superior or interior screw angulation to the severity of the mandibular ramus height loss. Only the anterior screw insertion angle reduced the amount of ramus height loss (CC = −0.72; R^2^ = 51%; *p* < 0.05). In this study, this angle varied from 96.5 degrees to 130 degrees (Figure 7).

A total of 21 cases of superior screw protrusion were found in this study; the range of the protrusion was from 1 mm to 5.8 mm. A total of 29 cases of medial protrusion were noted for the inferior screw position with a range of 1.00–8.32 mm. Only five cases of medial protrusion were found for the anterior screw with a range of 2.25–2.50 mm. A relatively weak relationship between the superior screw protrusion and ramus height loss was found (CC = −0.28; R^2^ = 8%; *p* < 0.05). Such a weak relationship was observed for the inferior screw protrusion (CC = 0.38; R^2^ = 14%; *p* < 0.05). For anterior screw incidences of protrusion, the relation was moderately strong (CC = 0.94; R^2^ = 88%; *p* < 0.05). The collective results are presented in Figure 8.

When examining the quality of the union, it is important to note that the optical density at the consolidation site is higher than that of the surrounding bone 12 months after surgery. In addition, structural analysis of the image of this bone at the post-fracture site (BI), indicates a bone different from the cancellous bone (Table 2). Bone density at the post-fracture site is directly proportionally related to the Bone Index (CC = 0.27; R^2^ = 7%; *p* < 0.05). The value of the correlation coefficient indicates a relatively weak relationship between the variables.

Only the distance of superior screws from the fracture line is related to BI: the closer to the fracture line of the inserted screw, the closer the bone structure in the union is to the cancellous bone (CC = −0.28; R^2^ = 8%; *p* < 0.05).

In contrast, the insertion angle of the superior screw has no effect on the quality of bone union, unlike the inferior screw (CC = 0.34; R^2^ = 12%; *p* < 0.05) and anterior screw (CC = −0.78; R^2^ = 61%; *p* < 0.05), whose inclination is associated with BI values (Figure 9). The association of the inferior and anterior screw angulation with BI is the opposite. Higher angulations of the inferior screw (i.e., 120–130 degrees) favor high BI values. For the anterior screw, the best union qualities yield lower angles; i.e., a nearly perpendicular (95–109 degrees) insertion of this screw is associated with high BI values.

The amount of superior screw protrusion to the medial side of the mandible is directly proportionally weakly related to an increase in BI values (CC = 0.37; R^2^ = 14%; *p* < 0.05). The other two screw locations do not show a relationship of screw tip protrusion size with the Bone Index.

Assessing only type B fractures, a superior screw distance of 6.30 ± 2.83 mm from the fracture line and its moderately strong association with the loss of ramus height was noted (CC = 0.81, R^2^ = 65%, *p* < 0.05). The case is similar for the inferior screw (CC = 0.88, R^2^ = 77%, *p* < 0.05), where the distance was 9.42 ± 3.52 mm. The quality of consolidation as measured by the BI value in type B fractures was 0.62 ± 0.28. The study also noted that larger insertion angles of the superior screw were associated with an increasing loss of ramus height. This relationship is moderately strong (CC = 0.57, R2 = 33%, *p* < 0.05). The amount of ramus height loss was the same in type C fractures (3.93 ± 3.09 mm) than type B (4.07 ± 2.61 mm).

As far as only type C fractures were considered, BI was similar (0.67 ± 0.29), as with type B fractures. It was noted that there was a weak direct proportional association of the superior screw distance (3.20 ± 1.15 mm) with the amount of ramus height loss (CC = 40, R^2^ = 16%, *p* < 0.05) and a moderately strong inverse proportional association of anterior screw distance from the fracture line (5.46 ± 0.89 mm) with the amount of ramus height loss (CC = −0.71, R^2^ = 50%, *p* < 0.05). A moderately strong inverse relationship was observed for the size of the anterior screw insertion angle (118° ± 12°) and the amount of ramus height loss (CC = −0.72, R^2^ = 51%, *p* < 0.05). On the other hand, in type C fractures (as opposed to the absence of such a relationship in type B), an association of the protrusion of the screw to the inner side was noted. For protrusion in the superior screw (1.21 ± 1.96 mm), a weak inverse relationship with ramus height loss was noted (CC = 0.30, R^2^ = 9%, *p* < 0.05). For the inferior screw, a relatively weak direct relationship to ramus loss was also found (CC = 0.42, R^2^ = 17%, *p* < 0.05). A small number of fixing material protrusion was not allowed for the statistical analysis of the anterior screw protrusion with ramus loss.

Between one and four screws per osteosynthesis were used in the study group of patients. However, two screws or three pieces were most commonly used. Combining these subgroups separated patients with fixations with a smaller number of screws (one or two used) accounting for 61 fixations and a group treated with a larger number of screws (three or four screws) accounting for 13 osteosyntheses. The scanty-screw (3.81 ± 2.81 mm) vs. multi-screw (4.69 ± 3.61 mm) subgroups did not differ in the amount of ramus height loss or in the Bone Index, i.e., union quality (0.63 ± 0.29 mm vs. 0.78 ± 0.29 mm, respectively). A moderately strong direct proportional relationship was detected between the post-traumatic angulation of bone fragments and the final resultant loss of mandibular ramus height (CC = 0.51, R^2^ = 26%, *p* < 0.05). The primary post-traumatic overlap of the fragments (ramus shortening) was directly proportionally associated with the final treatment outcome (CC = 0.34, R^2^ = 12%, *p* < 0.05). However, this relationship was relatively weak.

## 4. Discussion

It is previously known that the number of screws used determines the stability of fixation. Using a single screw [29], significant mobility in the fracture gap is achieved, which can push already reduced bone fragments more than 500 µm away from each other [21]. If two or three screws are used then the displacement of bone fragments can be reduced to about 200 µm. Furthermore, it is known that the use of more screws (i.e., three) reduces the cumulative stress around the fixation.

This clinical study indicates that not only the number of screws but also the positioning of the screws is important for treatment success. With increasing distance of the superior screw insertion site from the fracture line, a reduction in mandibular height loss at the 12-month follow-up can be observed in the clinical material presented here. It can be stated that placing the superior screw too close (i.e., 1–2 mm below the fracture) in relation to the fracture line leads to a large loss of ramus height. It can be assumed that the mechanical fixation of the screw in the distal fragment (i.e., ramus) is too weak and that masticatory forces can cause a gradual downward tilt of the medial pole of the mandibular head. On the other hand, an inverse relationship can be observed with regard to the influence of the location of the interior screw. The closer the inferior screw is to the fracture line, the better the long-term outcome (less loss of ramus height). The regression lines intersect at a point corresponding to a distance of approximately 6.5 mm from the fracture line. The superior and inferior screw cannot be inserted in the same place. The observation of the two regression lines reveals a flattening of the regression lines, which allows indicating the optimal distance for each screw. For the superior screw, the optimal insertion site is about 4 mm below the fracture line, while for the inferior screw, the optimal insertion site is 7–8 mm below the fracture line. A guideline from these studies for the location of the anterior screw is that it is best placed 4–5 mm below the fracture line. Sometimes only two screws are possible to place [30]. It would be to place only the superior and inferior screw.

In light of these studies, the role of the anterior screw seems important in the osteosynthesis of the mandibular head. If there is room for maneuver, better long-term anatomical results will be obtained when the screw insertion angle is close to 130 deg. (i.e high) in the coronal plane (and worse results when it is closer to 96 deg.). As far as the angulation of the other screws is concerned, the operator is free to act and can be guided by the assessment of the intraoperative situation in choosing the insertion angle of the superior and inferior screws. All angles from the range 90°–140° can be utilized without any negative influence on later mandible ramus height.

The relationship between the amount of protrusion of the superior screw and the amount of loss of the mandibular ramus height is opposite to that of the inferior and anterior screws. The amount of protrusion of the latter two screws on the inner side of the mandibular condylar process is directly proportionally related to the amount of ramus height loss after 12 months (especially the anterior screw). This seems easy to explain. The more the screw protrudes on the medial side of the mandible (or rather, in the pterygoid fovea direction for the anterior screw protrusion), the more it damages, e.g., presses on tissues that supply blood to the mandibular head, i.e., the lateral pterygoid muscle [31]. Surprisingly, the size of the superior screw protrusion is inversely proportional to the size of the loss of the mandibular ramus (a similar astonishing relationship was detected when analyzing the quality of the union on the basis of BI), as if bigger protrusion were healthier. However, this is probably not the case, and the part of the screw protruding beyond the bone certainly has no positive role in the mechanics of osteosynthesis. It is impossible to exploit the long screw advantage [17,32]. Another protrusion effect for the superior screw may result from the fact that some of these perforations affect the articulation surface, which significantly disturbs the joint function. Even after a quick removal of the screw, the healing of such an osteochondral wound may be complicated. In general, the matter requires further investigation and it is obvious that going through-and-through the mandibular head during osteosynthesis should be avoided. The Bone Index is a known measure of whether or not the examined bone has the characteristics of a valid cancellous bone [22,23,33]. To reconstruct a cancellous bone in a union place, the BI value would be expected to be approx. 0.9, which is actually much less in this clinical study. Thus, this is how the bone differs significantly from the appearance of the trabecular bone (*p* < 0.05). However, there is a tendency that the closer the superior screws are, the higher is the BI union. If one assumes that the appearance of the bone at the fracture site is different for years than the intact bone [34], one could assume that low BI values at the post-fracture site are desirable and natural. This would be the second indication resulting from this work to insert the superior screw as low as possible. In terms of the quality of the bone union, the location of the other screws does not matter.

Thus, it seems that not only the screw diameter [19], length [35], thread construction [36], number of screws [37], and mechanical properties [38], but the smart placement of them in the mandibular head, leads to therapeutic success.

A weakness of the study is the difficulty in assessing the role of the anterior screw due to the few patients fixated by three screws (this is a requirement for the existence of a screw in the anterior position). Despite the inclusion of 57 patients and 74 fractures, this material needs to be enlarged and especially continued in a multicenter dimension.

## 5. Conclusions

Based on the collected clinical material and the analysis of the mandibular ramus height loss at 12 months after surgery, it can be noted that the optimal location for the superior screw after the reduction of a mandibular head fracture appears to be approx. 4 mm below the fracture line (with any angulation), together with the inferior screw approx. 8 mm (with any angulation), and anterior screw position approx. 4–5 mm distant from fracture line, with the best angulation of 130 degrees in the coronal plane. The use of wrong angles and too long screws lead to the protrusion of the screw tip on the mesial/anterior side of the mandible head and may worsen the long-term results of the treatment of head fractures.

## Figures and Tables

**Figure 1 jcm-11-02031-f001:**
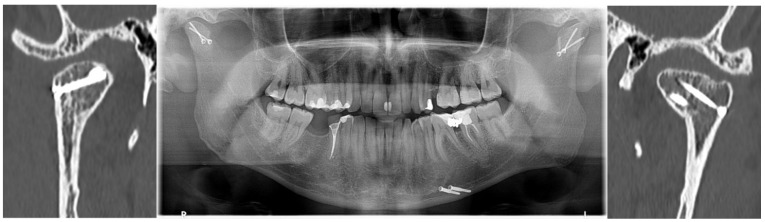
Radiological documentation showing the effect of surgical treatment of mandibular head fractures (examinations were acquired 12 months after osteosynthesis). A pantomogram radiograph is shown in the center as an overview study and CT coronal sections through the mandibular heads are shown on the sides. In the CT scan, the union and position of the headless compression screws can be accurately assessed (volumetric study makes it possible to obtain any cross-section). This patient had an additional fracture in the rim of the mandibular body on the left side. It was treated with lag-screws. Please note the small depression in the articular surface of the left mandibular head (CT image on the right side of this figure). It appears to be the result of a tiny shift between the fixated bone fragments. This shift, however, changes the height of the mandibular ramus.

**Figure 2 jcm-11-02031-f002:**
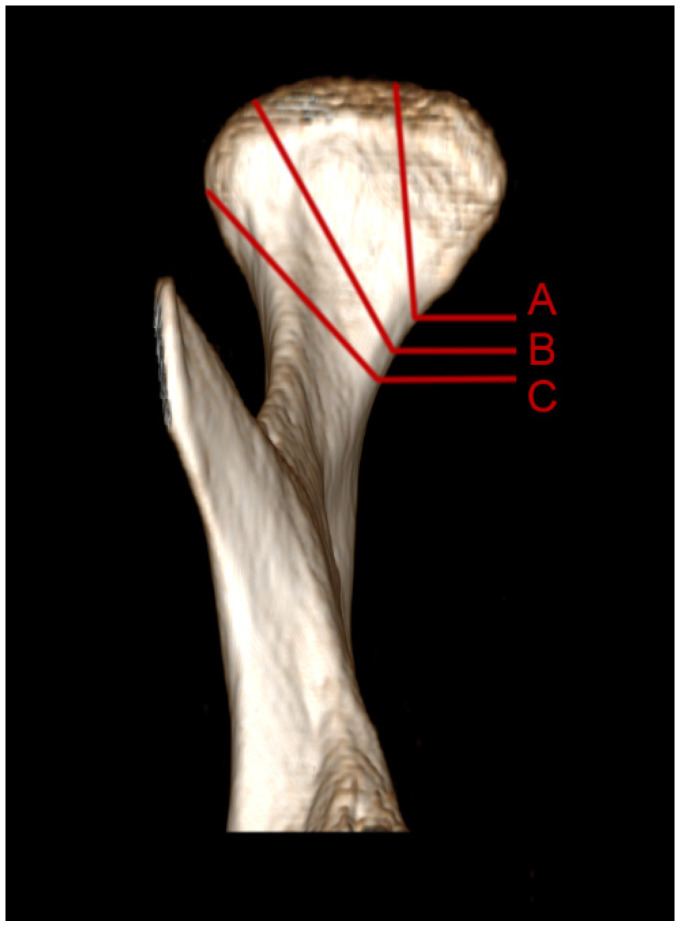
Illustration of mandibular head classification based on the right mandibular condylar process [19]. A—fracture of the medial pole of the mandibular head. B—the fracture line runs through the lateral pole of the mandibular head or near its medial portion. C—the fracture line runs just below the lateral pole of the mandibular head.

**Figure 3 jcm-11-02031-f003:**
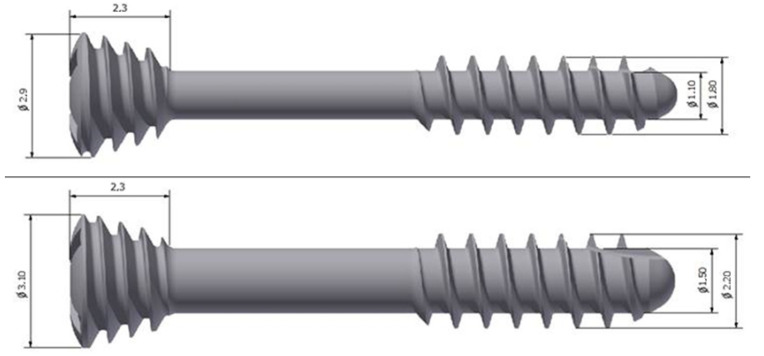
Two types of headless compression screws were used in this study: 1.8 titanium and 2.2 magnesium by ChM.

**Figure 4 jcm-11-02031-f004:**
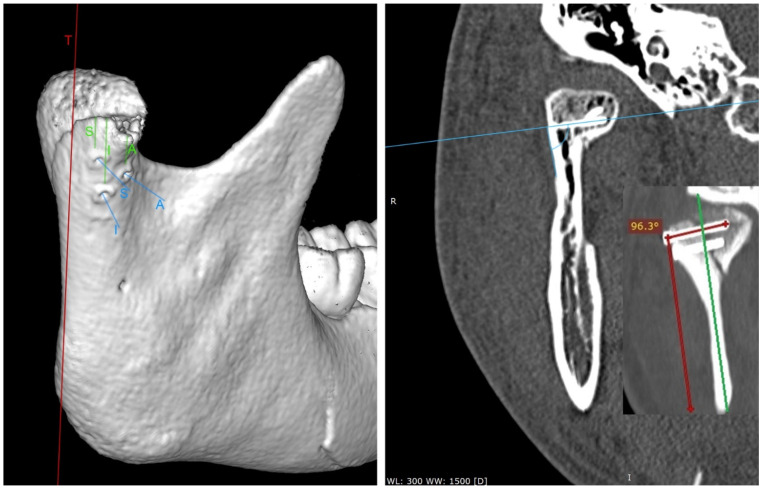
Postoperative radiological examination. On the left, computed tomography acquired immediately after the surgery (view ¾ laterally). This 3D imaging reconstruction shows the screw location used in the study. T—tangent line in the lateral view to the posterior edge of the mandible. The measurements of the distance between the screws and the fracture line are marked in green. The screw insertion directions are shown in blue. S—superior screw; I—inferior screw; A—anterior screw. On the right, a coronary cross-section (taken 12 months after the end of treatment) set on sample screws and measures their angle of insertion into the bone.

**Figure 5 jcm-11-02031-f005:**
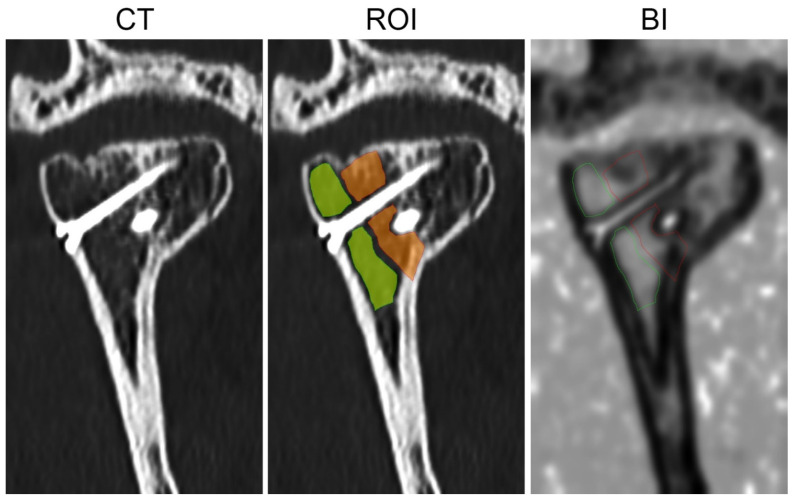
The method of bone union structure evaluation. CT—coronal section of mandible head in examination taken 12 months post-operationally. ROI—orange region covered post-fracture line area; green region describes control site of cancellous bone. BI—map of the Body Index, in which lighter regions indicate high value of the texture feature contrary to dark regions, where BI has low value. BI, as used in skeletal tissue image, is the marker of normal cancellous bone.

**Figure 6 jcm-11-02031-f006:**
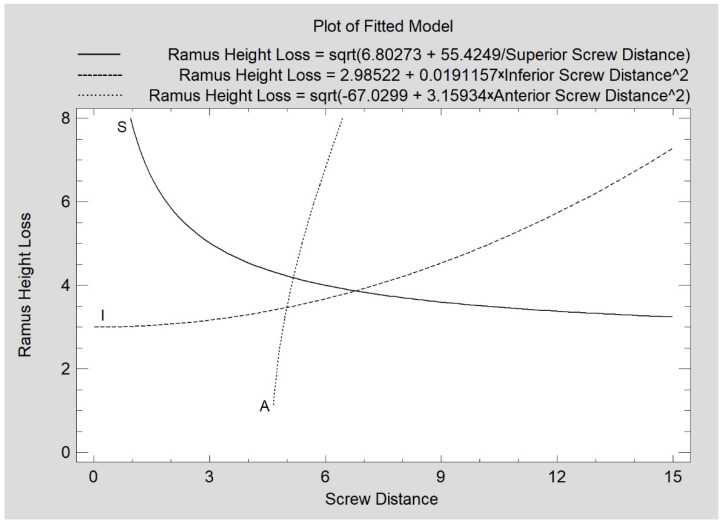
Relationship of mandibular ramus height loss (mm) measured 12 months post-operationally with the distance of the inserted screw from the fracture line (mm). All relationships are statistically significant (*p* < 0.05). S—superior screw; I—inferior screw; A—anterior screw.

**Figure 7 jcm-11-02031-f007:**
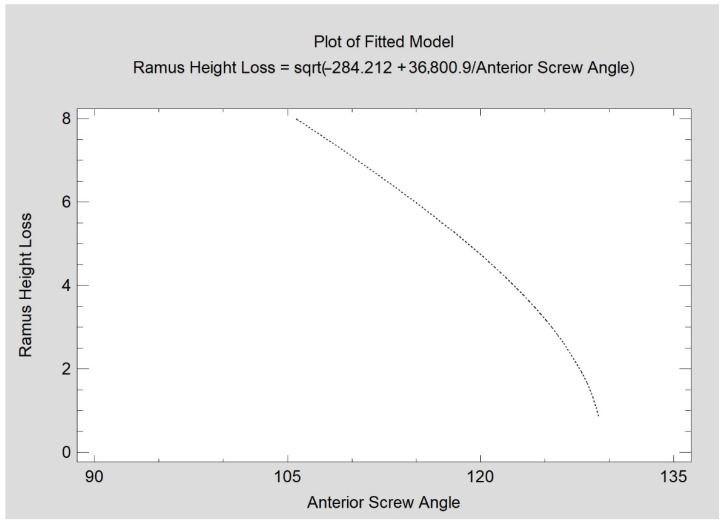
Relationship of mandibular ramus height loss (mm) measured 12 months post-operationally with angulation of the anterior fixing screw (degrees). The relationship is statistically significant (*p* < 0.05).

**Figure 8 jcm-11-02031-f008:**
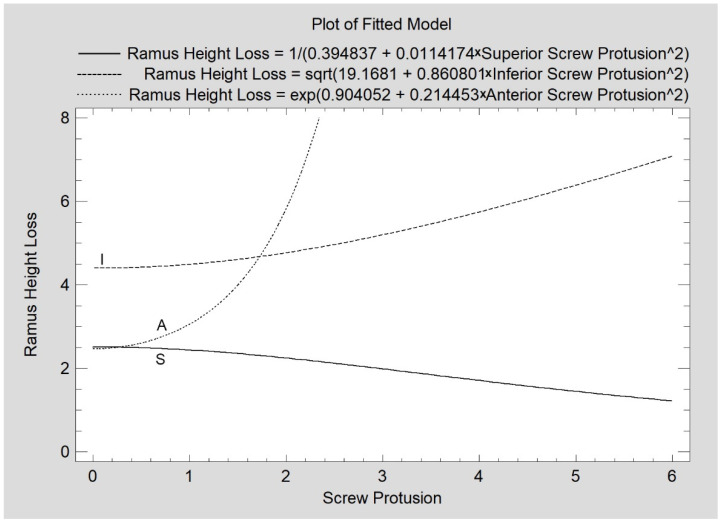
Influence of screw protrusion at the time of surgery on the fate of maintaining the height of the mandibular ramus. The values of protrusion for each screw location are statistically significantly related to the changes in the ramus height (*p* < 0.05). S—superior screw; I—inferior screw; A—anterior screw.

**Figure 9 jcm-11-02031-f009:**
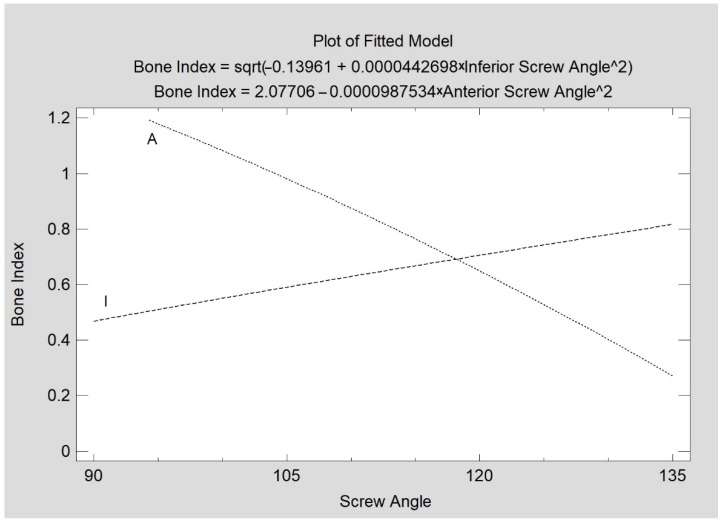
The relationship between the consolidation quality (Bone Index) and the screw insertion angle was observed. Statistically significant relations (*p* < 0.05) were found for the inferior screw (I) and for the anterior screw (A).

**Table 1 jcm-11-02031-t001:** Data for fixation screws inserted into the mandibular head.

Fixation Screw Position	Distance from Fracture	Angulation in Frontal Plain	Protrusion of the Tip
Superior screw (S)	4.00 ± 2.19 mm	107^0^ ± 15^0^	0.96 ± 1.77 mm
Interior screw (I)	7.11 ± 3.08 mm	118^0^ ± 11^0^	1.50 ± 2.31 mm
Anterior screw (A)	5.46 ± 0.89 mm	118^0^ ± 12^0^	0.73 ± 1.13 mm

**Table 2 jcm-11-02031-t002:** Evaluation of bone union quality after twelve months of healing and remodeling.

Feature	ROI	Value	Note
Bone density	Post-fracture site	457 ± 118 HU	*p* < 0.05
	Control site	388 ± 91 HU	
Bone index (BI)	Post-fracture site	0.66 ± 0.29	*p* < 0.05
	Control site	0.90 ± 0.30	

## Data Availability

The data presented in this study are available on request from the corresponding author. The data are not publicly available due to an ongoing multicenter project.

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
