# Peer review of "The Osteosynthesis of the Mandibular Head, Does the Way the Screws Are Positioned Matter?"

_jcm, 2022, doi:10.3390/jcm11072031_

Round 1
Reviewer 1 Report
Thank you for allowing me to evaluate this manuscript. I have evaluated the study " Mandibular head osteosynthesis, does the way the screws are positioned matter? " I think overall, the paper is interesting. However, there are some concerns that I would like to address and share with the authors.
-The title can be changed as follows. So it can be more understandable.
“The osteosynthesis of the mandibular condyle, does the way the screws are positioned matter?”
-A figure can be added regarding the neff classification specified in the material method -section, line 67.
- Patients operated received titanium fixation and magnesium fixation (the choice of fixation material was decided on the day the patient arrived at the hospital). In the conclusion part of the article, only a comparison of the positions of the screws has been made. so why not use one type of fixing screw?
- Is there any difference between titanium fixation and magnesium fixation. Mechanical properties of a material refer to the response of a material to external stress. A material’s mechanical properties can be measured by applying macro-, micro-, and advanced nano-testing and standards.? The following article describing this situation can be added to the references. “Altan, Ahmet, İbrahim Damlar, and Osman Åžahin. Can resorbable fixation screws replace titanium fixation screws? A nano-indentation study. Journal of Oral and Maxillofacial Surgery 74.7 (2016): 1421-e1.”
Author Response
Please, find the attached file with detailed explanation.

Reviewer 2 Report
- Interesting study on a very important topic
- 3 screw for fixation of condylar head was the nice concept, loss of mandibular height is the concern although fixation with 3 screws.
- Small sample size and limited follow up are the short coming of the study.
Author Response

(The authors gave the same response as above.)

Reviewer 3 Report
Thank you for your study looking at open reduction and internal fixation of condylar head fractures. these remain a challenging problem to manage, either non surgically or with surgery. I do have a number of questions and comments:
Over what period was the study carried out?
Was there any difference between titanium and magnesium screws in terms of outcome?
What is meant by internal medicine patients? The severity of illness can vary widely with significantly different implications on recovery and outcomes.
The aim of management of mandibular fractures is primarily restoration of pre-injury occlusion and maintenance of good mouth opening. Ramus height and bone index are secondary outcomes and not necessarily clinically relevant. Although you were able to demonstrate a relationship with change in ramus height, it would be of much more relevance to look at the clinical outcomes in these patients and whether there was any difference.
Author Response

(The authors gave the same response as above.)

Round 2
Reviewer 3 Report
Thank you for your response to my previous comments. Given that this paper is one of a series looking at this problem, it makes sense for it to be published in its present form.